# Deep Learning Approaches to Earth Observation Change Detection

**Antonio Di Pilato** [1,*,†] **, Nicolò Taggio** [2], **Alexis Pompili** [1], **Michele Iacobellis** [2], **Adriano Di Florio** [3], **Davide Passarelli** [2] **and Sergio Samarelli** [2]

1 Dipartimento Interateneo di Fisica, Università Degli Studi di Bari Aldo Moro, 70126 Bari, Italy; alexis.pompili@uniba.it
2 Planetek Italia s.r.l, 70132 Bari, Italy; taggio@planetek.it (N.T.); iacobellis@planetek.it (M.I.); passarelli@planetek.it (D.P.); samarelli@planetek.it (S.S.)
3 Dipartimento Interateneo di Fisica, Politecnico di Bari, 70126 Bari, Italy; adriano.diflorio@poliba.it
* Correspondence: antonio.dipilato@uniba.it
† Current affiliation: Center for Advanced Systems Understanding (CASUS), 02826 Görlitz, Germany.

**Abstract:** The interest in change detection in the field of remote sensing has increased in the last few years. Searching for changes in satellite images has many useful applications, ranging from land cover and land use analysis to anomaly detection. In particular, urban change detection provides an efficient tool to study urban spread and growth through several years of observation. At the same time, change detection is often a computationally challenging and time-consuming task; therefore, a standard approach with manual detection of the elements of interest by experts in the domain of Earth Observation needs to be replaced by innovative methods that can guarantee optimal results with unquestionable value and within reasonable time. In this paper, we present two different approaches to change detection (semantic segmentation and classification) that both exploit convolutional neural networks to address these particular needs, which can be further refined and used in post-processing workflows for a large variety of applications.

**Keywords:** change detection; convolutional neural network; earth observation; deep learning; Sentinel-2

## 1. Introduction

Change detection in the field of remote sensing is the process of identifying differences in the state of an object or phenomenon by observing it at different subsequent times [1]. The nature of this task thus requires the usage of methods and algorithms that compare two or more satellite images of the same scene and produce results that can either map changes pixel by pixel or provide an overall classification of the input data as including or not relevant changes within the analyzed area.

Specifically, change detection applied to urban areas [2,3] is an interesting case study as it aims at monitoring differences in land cover due to urban expansion and spread over the years in a more automated way. Although a large variety of methods exists and has been classified and discussed [4–6], and classical methods based on pixel-difference vectors are constantly developed and studied [7], the deep learning approach to this problem is currently object of active research and ranges from the design of innovative methods exploiting conditional generative adversarial networks (cGANs) [8] to more complex methods that obtain fine change detection results starting from coarse scene classification [9]. With the advent of GPUs and heterogeneous computing, the computational load of deep learning models for Earth Observation is not prohibitive as in the past, due to the considerable reduction of training time. This scenario allows inspecting the performance of several algorithms in a more efficient way, without dealing with tight restrictions. In particular, convolutional neural networks (CNNs) can easily exploit the different architecture

of these devices with respect to traditional CPUs and allow achieving valuable results in shorter time.

At the same time, the training of models with a high level of generalization capabilities is extremely challenging, due to the changing urban patterns from a country to another. In addition, vegetation and seasonal changes often represent a source of contamination that cannot be easily removed, and contribute to lower the performance of efficient convolutional neural networks.

In this scenario, performing change detection with CNNs has the objective of providing a fast and useful analysis of an area that can be integrated into traditional post-processing workflows on the ground stations, which combine results from several algorithms to obtain a final map of all the changes present in the scene. This objective can be achieved with two different approaches: semantic segmentation and classification. The first approach is generally suitable for the full analysis on ground: the acquired data are downloaded at the ground facilities and the deep learning algorithm provides a detailed map of all the changes detected within the area. The second approach is onboard-oriented, as it aims to evaluate the change content of a scene and transmit such estimation to the ground stations; the latter can further decide whether to download or not the new acquired data for further analyses. Specific monitoring tasks can be planned by exploiting the onboard processing to filter the interesting data only, thus mitigating the download bottleneck.

The aim of this study is to provide high performance algorithms that can be exploited by both private companies (which usually have more complex change detection workflows) and academic institutions (for research studies). With this purpose in mind, only free-access tools have been used, as well as public data collected by the Sentinel-2 satellites.

## 2. The Sentinel-2 Mission

The Sentinel-2 mission consists of a constellation of two twin satellites flying in the same Sun-synchronous polar orbit but phased at 180° at a mean altitude of 786 km, designed to give a high revisit frequency of five days at the Equator. It aims at monitoring the variability of the land surface conditions through the acquisition of high resolution multispectral images that can be exploited for land cover/change classification, atmospheric correction and cloud/snow separation.

Each satellite carries a multispectral instrument (MSI) for data acquisition [10]. The MSI measures the Earth's reflected radiance in 13 spectral bands: the visible and near-infrared (VNIR), and the short-wave infrared (SWIR), with three different spatial resolutions (10 m, 20 m, and 60 m). Additional information is provided in Table 1.

**Table 1.** Spectral bands of the Sentinel-2 twin satellites' sensors [10].

| | S2A | | S2B | | |
| :---: | :---: | :---: | :---: | :---: | :---: |
| Band Number | Central Wavelength (nm) | Bandwidth (nm) | Central Wavelength (nm) | Bandwidth (nm) | Spatial Resolution (m) |
| 1 | 442.7 | 21 | 442.3 | 21 | 60 |
| 2 | 492.4 | 66 | 492.1 | 66 | 10 |
| 3 | 559.8 | 36 | 559.0 | 36 | 10 |
| 4 | 664.6 | 31 | 665.0 | 31 | 10 |
| 5 | 704.1 | 15 | 703.8 | 16 | 20 |
| 6 | 740.5 | 15 | 739.1 | 15 | 20 |
| 7 | 782.8 | 20 | 779.7 | 20 | 20 |
| 8 | 832.8 | 106 | 833.0 | 106 | 10 |
| 8a | 864.7 | 21 | 864.0 | 22 | 20 |
| 9 | 945.1 | 20 | 943.2 | 21 | 60 |
| 10 | 1373.5 | 31 | 1376.9 | 30 | 60 |
| 11 | 1613.7 | 91 | 1610.4 | 94 | 20 |
| 12 | 2202.4 | 175 | 2185.7 | 185 | 20 |

Sentinel-2 data are downloaded at the ground segment, which uses several processing algorithms to obtain two types of final products, in the form of compilations of elementary granules or tiles (minimum indivisible partitions) of fixed size acquired within a single orbit. The Level-1C (L1C) products are composed by $100 \times 100$ km$^2$ tiles, ortho-images in UTM/WGS84 projection. Pixel values of L1C products are provided as top-of-atmosphere (TOA) reflectance measurements, along with the parameters to transform them into radiance measurements. Cloud masks and information about ozone, water vapour, and mean sea level pressure are also included. The Level-2A (L2A) products consist of bottom-of-atmosphere (BOA) reflectance images derived from the corresponding L1C products by means of atmospheric correction algorithms. Therefore, each product is composed by $100 \times 100$ km$^2$ tiles in cartographic geometry (UTM/WGS84 projection). L2A products are generated at the ground segment since 2018, but the L2A processing method can be applied by users to even older data through the Sentinel-2 Toolbox.

### 2.1. Change Detection Dataset

A common limitation of using deep learning algorithms for change detection is the poor availability of already labeled datasets. Despite large amounts of satellite data are acquired and downloaded daily, the process of assigning ground truths to images in any format (depending on the specific task) is a well-known time-consuming operation. In this study, the Onera Satellite Change Detection (OSCD) dataset [11] was used for the purpose. It consists of 24 Sentinel-2 L1C image pairs, each capturing an area of approximately $600 \times 600$ pixels at 10 m resolution; bands at 20 m and 60 m resolution of the L1C products were upsampled to the resolution of 10 m such that all the channels have aligned pixels. Even though the ground truths of all the 24 image pairs are currently available, only 14 regions were used for training purpose, as originally suggested by the authors of the dataset. In addition, ground truths are provided in the form of binary change maps where each pixel has a value marking if a change occurred at that location within the original image pair or not.

Images of each pair were taken at a temporal distance of 2–3 years; a very small amount of clouds is present in some of them and no restrictions were set on brightness conditions, as vegetation, seasonal, and sunlight exposure changes were ignored. A sample image pair (RGB-only) and its corresponding change map are shown in Figure 1.

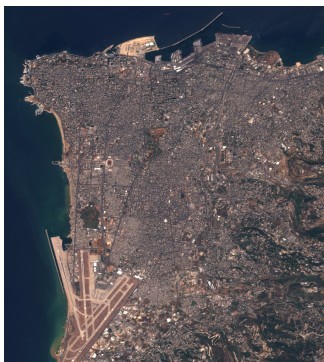 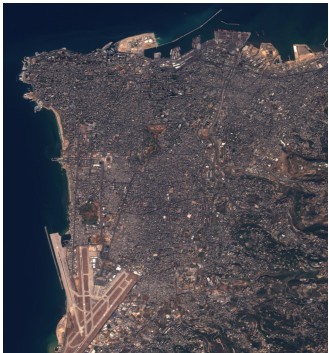 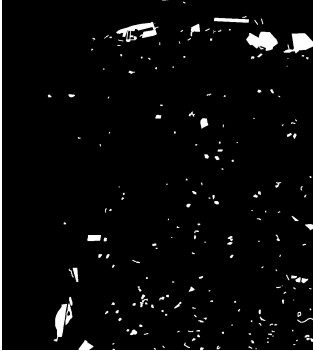

**Figure 1.** Sample image pair and corresponding ground truth of the Beirut area. Left is the image before changes (20 August 2015), center is the image after changes occurred (3 January 2017), and right is the change map based on ground truth. White pixels map the changes detected in the scene [11].

### 2.2. Dataset Splitting Strategy

To have a rich and large training dataset for the purpose of the study, patches with size of $128 \times 128 \times 13$ pixels were extracted from the original 14 image pairs according to the following method. While 13 corresponds to the number of the spectral bands available for Sentinel-2 images, the size of 128 was selected as the best compromise between the optimal choice for fully convolutional neural networks (128 to 256) and the need of limiting

the number of overlapping pixels of different patches extracted from the same original image (as described below).

First, a random pixel location $(x, y)$ was selected within the original pre-changes image, and the area of pixels $[x, y, x + a, y + a]$, with $a = 128$, was cropped. Such selection was operated while having the cropped patch entirely confined in the scene borders. The same crop was extracted from the corresponding post-changes image and the change map. Then, the ratio $R$ between the number of changed pixels and the total number of pixels for the selected patch was evaluated on the binary change map and two possible cases established:

1.  $0 \leq R < 0.1$ (changes cover less than 10% of the area captured by the patch);
2.  $R \geq 0.1$ (the 10% or more of the area captured by the patch is covered by change pixels).

In the first case, a random transformation among six possible choices (rotation by 90°, 180°, 270° or 360°, vertical or horizontal flip) was applied to the two images of the pair and to the corresponding change map, and the results were stored in the training dataset. In the second case, instead, all the six available transformations were operated, one per time, and the resulting patches and change maps saved and stored as well. A total of 500 crops were randomly selected for each of the 14 regions; possible duplicates were removed and the final training dataset consisted of 13919 input pairs and ground truths.

It should be observed that the randomness introduced through the crop selection and the applied transformation(s) reduces the effect of pixel correlation due to patches being extracted from the same original region. Indeed, patches do not share a fixed number of pixels which might heavily bias the optimization step during the training process. Furthermore, those patches that share a certain amount of pixels can be still distinguished by a transformation, enhancing such reduction effect.

Finally, while for the semantic segmentation approach the ground truths are originally available in the OSCD dataset, for the classification task they were generated by applying the following criterion: if the changed pixels of the $128 \times 128$ ground truth are less than 25, then the image pair is labeled as 0 (not interesting for further analysis on ground); otherwise, it is labeled as 1. This particular choice was made according to the idea that we are more interested in detecting relevant changes at the cost of a higher fraction of false positives rather than losing important information (it is up to the analysis on ground to use post-processing algorithms and discard fake changes).

## 3. Model Architectures

The design of the model architectures is highly connected to the approach used for change detection. As a consequence, the models developed for semantic segmentation cannot be used to also perform classification, due to different output formats. In the semantic segmentation approach, a UNet-like architecture [12] was designed, as it is proven to be a very efficient kind of DL algorithm to provide results with the required dimensionality. It is essentially composed by two symmetrical CNNs: the first part extracts features from the input data through a series of convolutions and pooling operations, while in the second part transpose convolutional layers are used together with the skip-connection technique to recover spatial information from the previous layers and use them at new abstraction levels. The final output is obtained by a convolutional layer with a $1 \times 1$ kernel and sigmoid activation. For the classification method, a traditional CNN was used: after several convolutional and pooling layers, a flattening operation is performed to reduce the dimensionality of the processed data to 1, and a few dense layers with sigmoid activation are used before the 1-neuron sigmoid output layer, which provides the overall score indicating whether the image pair contains or not a significative amount of changes.

The main element of the neural networks that were designed for both the approaches is the convolutional unit. It has been shown that two or more consecutive convolutional layers with a small kernel (i.e., $3 \times 3$) has the effect to enlarge the receptive field (the "window" from which features are extracted by a kernel) while keeping the number of parameters smaller with respect to using a single convolutional layer with a larger kernel [13].

Therefore, the basic convolutional unit used in all the models is composed by two stacked convolutional layers, each of them followed by a batch normalization layer that allows to obtain better convergence while speeding up the training process [14]. Rectified linear unit (ReLU) was selected as the activation function of the convolutional layers, while a dropout layer [15] helps to reduce the overfitting effect and increase the model capacity. The logical structure of the convolutional unit is shown in Figure 2.

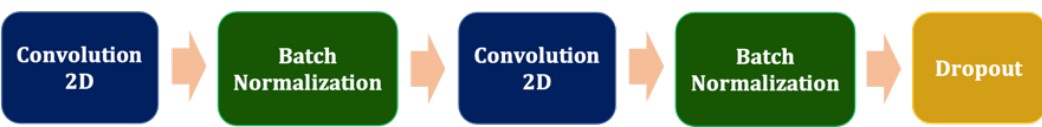

**Figure 2.** Basic convolutional unit used in all the model architectures.

Another important step of the architecture design process was the choice of how the network is going to compare the two images in each pair to produce the corresponding output. Two possible methods were evaluated: the *EarlyFusion* (EF) and the *Siamese* (Siam) methods. In the EF method, the two images of a pair are concatenated along the channel dimension as the initial operation of the neural network; features are then extracted from a single image with size $128 \times 128 \times 26$. In the Siam method, instead, feature extraction is operated on each of the two images separately but using shared weights between the two paths created in the network, while the skip-connection steps recover the spatial information from both the processed images at different abstraction levels.

Generally, the *Siamese* method makes both the training and inference processes slower, because the concurrent feature extraction doubles the size of the model in the first part of the network, while this does not occur with the *EarlyFusion* method. Indeed, if 16 different kernels are used in the first level of the neural network, 16 feature maps are thus produced from each of the two images, for a total of 32 feature maps, whereas in the common feature extraction of EF only 16 feature maps are produced in total. Moreover, a CNN cannot fully exploit the concurrent feature extraction part of the Siam UNet, because the skip-connection technique is not used (the classification network does not recover the spatial information from previous layers, as it is not needed to produce the single-score output). Therefore, as the classification approach mainly aims at achieving very fast execution in the onboard scenario, only the EF method was adopted for this task. The final architectures for classification and semantic segmentation are shown in Figure 3.

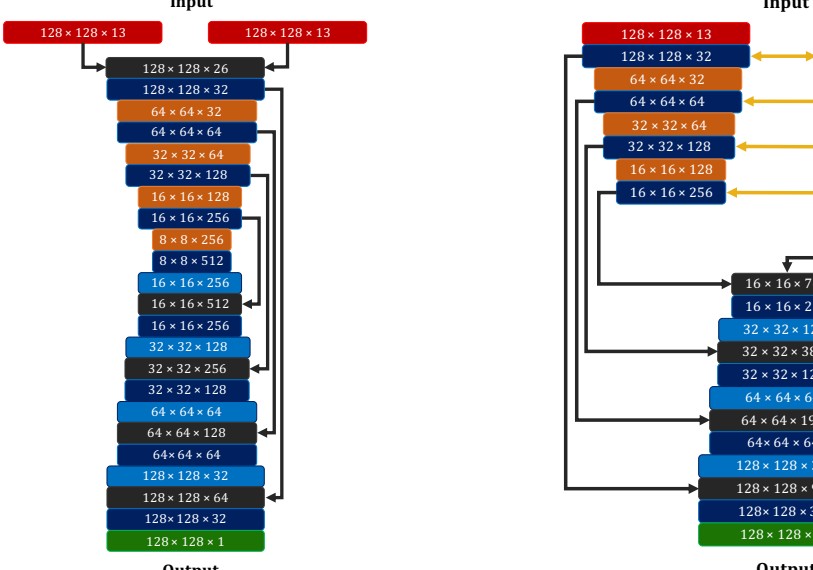

**Figure 3.** *Cont.*

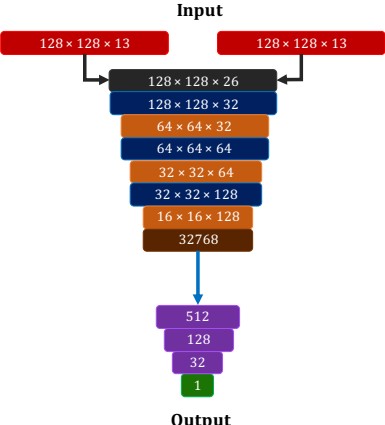

**Figure 3.** (**Top**) Model architectures used in the change detection task with the segmentation approach. The model on the left adopts the *EarlyFusion* method, while the model on the right uses the *Siamese* method. (**Bottom**) Model architecture used in the change detection task with the classification approach, adopting the *EarlyFusion* method only. Numbers indicate the output size of each layer of the network. Red is for inputs, dark blue for convolutional units, orange for max pooling layers, light blue for transpose convolutional layers, dark gray for concatenation layers, brown for the flatten layer, violet for dense layers, and green for output. Dark gray arrows indicate the concatenation operation, yellow arrows the shared weights and azure arrow corresponds to the batch normalization operation after flattening.

## 4. Training Details

The training process was designed having in mind that the dataset is poor and extremely unbalanced with respect to the class population and therefore the choice of the loss function, which must be minimized to reduce the error of the model predictions, was crucial.

The unbalance problem is a distinctive feature of change detection tasks. Indeed, changes can be considered rare events with respect to the pixel content of a satellite image, especially with a resolution of 10 m, where most of the changes involve very small areas (of the order of a few pixels). In the case of the OSCD dataset, the ratio between the number of changed pixels and the total number of pixels is extremely small (approximately 2%); therefore, any classifier trained without using methods to handle the unbalance between class samples would not be able to recognize changed pixels and most probably classify every pixel as unchanged, achieving apparently high performance.

Two solutions exist to solve such unbalance problem: (a) balancing the dataset by removing some samples from the most populated class and having more or less the same number of samples for both the classes, or (b) assigning weights to samples depending on their class population, during the minimization process of the loss function. The latter solution was adopted because the former method is not suitable for change detection. Indeed, the model needs to be trained also on image pairs which do not contain any change; otherwise, the algorithm will be biased and find at least the 2% of changed pixels within new scenes.

Two loss functions were used in this study. The first, denoted with "bce", is the binary cross-entropy function with the introduction of weights that are inversely proportional to the class population. For a batch of $N$ image pairs extracted from the training dataset, each having $M$ pixels, the loss function to minimize at each iteration is

$$\mathcal{L}^{bce} = -\frac{1}{N \times M} \sum_{i=1}^{N \times M} w_0 t_{0i} \log s_{0i} + w_1 (1 - t_{0i}) \log (1 - s_{0i}) \tag{1}$$

where $i$ denotes the pixel, $w_0$ and $w_1$ are the weights for the two different classes, $t_0$ and $t_1$ are the target labels at the two classes, and $s_0$ and $s_1$ are the predictions at the two classes.

Both the target labels and the predictions are used in the form of one-hot vectors (arrays in which each element corresponds to the probability of the sample to belong to each class).

The second loss function tested, denoted with "wbced", is the sum of the loss function of Equation (1) and the dice function

$$\mathcal{L}^{dice} = 1 - \frac{2y_p y_t + 1}{y_p + y_t + 1} \tag{2}$$

where $y_p$ is the predicted class and $y_t$ the ground truth, evaluated with binary method (1 for change, 0 otherwise). Dice loss is often adopted for segmentation tasks, as the dice coefficient (the fraction of Equation (2)) is a helpful metric to evaluate the similarity of contour regions [16] and to take into account the distribution of changed pixels across the change map, which is not uniform. This loss function was inspired by the study made with the UNet++ model for very high-resolution satellite images [17].

While both the considered loss functions are used for the segmentation approach, only the bce loss function is adopted for the classification task (without summing and averaging over the number of pixels *M*). In addition, the k-fold cross validation strategy is adopted in both the approaches, as it represents an efficient method to evaluate models' performances when dealing with datasets characterized by poor statistics. In this study, k is set equal to 5. Additional training details are included in Table 2.

**Table 2.** Training details of the segmentation and classification approaches.

|  | Segmentation | Classification |
|---|---|---|
| input format | $128 \times 128 \times 26$ or (2) $128 \times 128 \times 13$ | $128 \times 128 \times 26$ |
| output format | binary change map ($128 \times 128 \times 1$) | single score (1) |
| loss function | bce or wbced | bce |
| batch size | 128 | 128 |
| kernel size | $3 \times 3$ | $3 \times 3$ |
| regularization | L2 | L2 |
| optimizer | Adam | Adam |
| learning rate | $10^{-4}$ | $10^{-4}$ |
| dropout rate | 0.25 | 0.25 |

## 5. Performance Study

Several metrics were considered to evaluate the performance of each model. Given that the change detection task is a binary classification problem (either pixel-based or patch-based), the changed samples are considered "positives", as they carry the information of interest, while the unchanged samples are considered "negatives". The following quite usual notation is adopted: $T_P$ for true positives, $T_N$ for true negatives, $F_P$ for false positives, and $F_N$ for false negatives.

The classical accuracy score (fraction of correctly classified samples with respect to the entire set of data available, independent of the class population) is not suitable for the purpose, as it would overestimate the goodness of the predictions due to the unbalance problem. Indeed, if the 98% of the data is labeled as unchanged, any classifier could get 98% accuracy with predicting each sample as belonging to that class (included the changed



samples), thus committing an error of 2% and having no capability to detect changes at all. Nevertheless, *balanced accuracy* can be defined as

$$balanced\ accuracy = \frac{1}{2}\left(\frac{T_P}{T_P + F_N} + \frac{T_N}{T_N + F_P}\right) \tag{3}$$

to represent the mean value of the *accuracy* scores evaluated on the two classes separately. Although *balanced accuracy* is a good metric to evaluate model performance for tasks with unbalanced datasets, in this study three additional scores were considered. The *precision*

$$precision = \frac{T_P}{T_P + F_P} \tag{4}$$

represents the fraction of positive predictions that are actually correct. The *recall* or *sensitivity*

$$recall = \frac{T_P}{T_P + F_N} \tag{5}$$

represents the fraction of true positives that are correctly identified. Finally, the *F1-score*

$$F1 = 2 \times \frac{precision \times recall}{precision + recall} \tag{6}$$

combines *precision* and *recall* to provide a robust and reliable metric.

In this study, the *recall* was selected as the target metric to maximize. Indeed, it is crucial to reduce the fraction of false negatives, namely, those changes that are wrongly predicted by a certain model and thus lost, but accepting a higher number of false positives that can be later rejected by additional post-processing algorithms. The other metrics were also considered to ensure that the models have reliable performance despite the choice of maximizing the *recall*.

In the following section, the results for the segmentation approach are presented, whereas those for the classification approach are discussed in the subsequent one.

### 5.1. Segmentation Results

As previously mentioned, two different architectures were studied (EarlyFusion and Siamese, see Figure 3) and two loss functions were tested during the training process. As a result, four different segmentation models are here presented and named with an "Architecture-loss" logic: EF-bce, Siam-bce, EF-wbced, and Siam-wbced.

The ROC curves are shown in Figure 4. Each plot presents the ROC curves obtained by the five models trained with the k-fold cross validation technique, for each architecture and loss function. These results show that models trained with the bce loss function achieve higher *AUC* (*Area Under Curve*) scores than models trained with wbced, despite the latter loss function takes into account additional features of the change detection problem (i.e., clusters of changed pixels). The ROC curves prove that the trained models have good overall performance and are useful metrics to choose the working point for a certain task. In this study, however, they were used to observe the trend of the true positive rate (or *recall*) with respect to the false positive rate across the five models. Specifically, anomalous performance is observed for model 4 of EF-bce (top left), where the *AUC* score is lower than the other models trained with the same architecture-loss combination. This effect indicates that a possible issue occurred during training, such as the loss function stuck in a local minimum or a problem related to the dataset partition. As this effect is not present for other architectures and loss functions, where the same partitions are used, the second hypothesis was discarded. This issue can be thus considered a statistical effect due to the training process and not a specific feature of the models or task configuration.

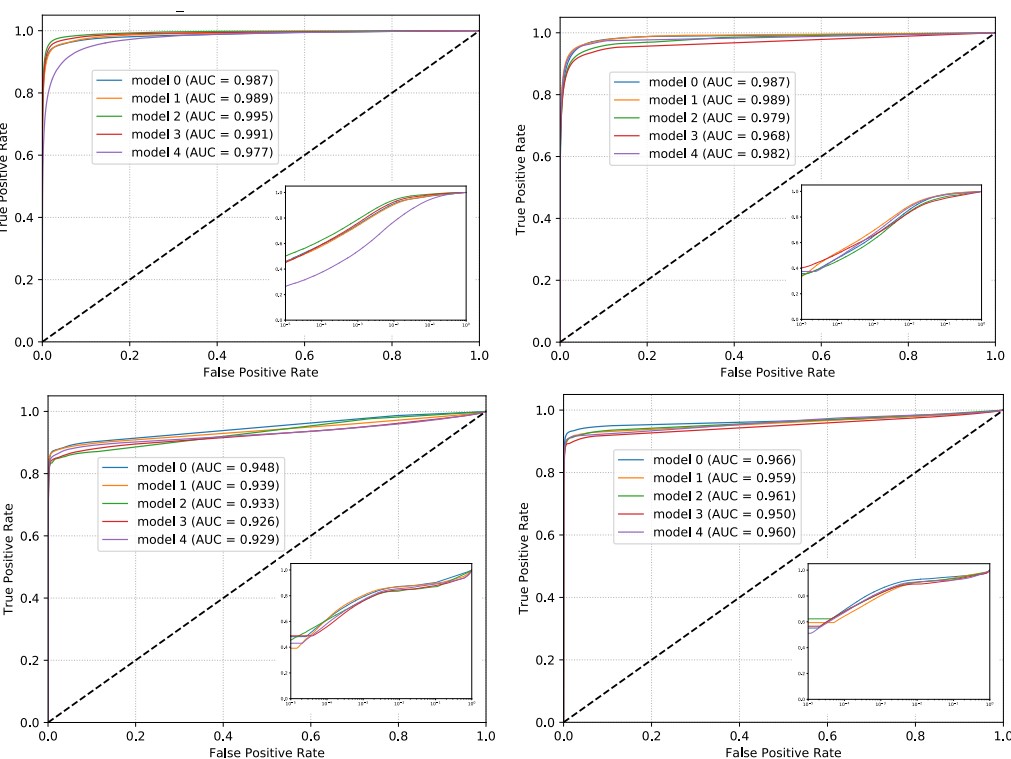

**Figure 4.** ROC curves of the five models trained with the cross validation technique for the segmentation task. **Top** are EF-bce (**left**) and Siam-bce (**right**) models, while **bottom** are EF-wbced (**left**) and Siam-wbced (**right**) models. The inset plots show the ROC curves with logarithmic scale for the *x*-axis.

The choice of the working point for each model was operated taking into account the average scores across the five trained model for different change thresholds. Indeed, the outputs of the segmentation models are change maps with each pixel having a value that ranges from 0 to 1. Setting a change threshold to 0.5 means that all the pixels above such value will be rounded to 1 (0 otherwise), but it does not necessarily represent the optimal choice for this study. As previously mentioned, the target performance to achieve in this work was to maximize the *recall* metric, while checking the other scores to ensure the reliability of the model performance. The corresponding value for the change threshold was found at 0.3 and chosen for all the segmentation models whose scores and results are presented in this section.

Figure 5 shows the confusion matrices obtained with the segmentation approach. As each model was trained on five different partitions of the dataset, only the best results are here presented, while detailed information about the average scores and their uncertainties, calculated as the standard deviations across the five models trained with the k-fold cross validation technique, are provided in Table 3. Models trained with the wbced loss function seem to have better overall performance, achieving *recall* values of 78.24% (EF-wbced) and 82.14% (Siam-wbced) and *F1-score* values larger than 85%. Furthermore, errors are smaller than 1%, thus indicating more stability during the training process with respect to the dataset partitions. On the other hand, models trained with the bce loss function have larger errors and lower *recall* values (75.26% and 73.93% for EF-bce and Siam-bce, respectively), but still achieve promising results.

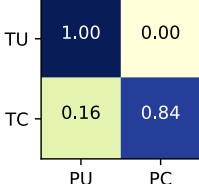 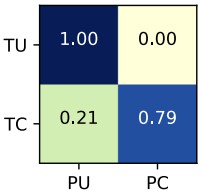 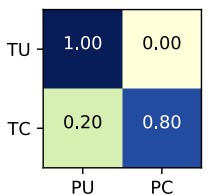 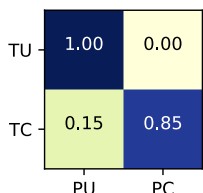

**Figure 5.** Confusion matrices obtained with the segmentation approach. The best results achieved with the k-fold cross validation technique are shown for (from left to right) EF-bce, Siam-bce, EF-wbced, and Siam-wbced methods. Rows are true unchanged (TU) and true changed (TC) classes (ground truths), whereas on columns are predicted unchanged (PU) and predicted changed (PC) classes (inference results).

**Table 3.** Average percentage values of the considered metrics for the segmentation task. The uncertainties are estimated as the standard deviations over the five models trained with the k-fold cross validation.

| Model | Precision | Recall | F1 | Balanced Accuracy |
|---|---|---|---|---|
| EF-bce | $91.45 \pm 1.68$ | $75.26 \pm 8.71$ | $82.33 \pm 5.96$ | $87.55 \pm 4.35$ |
| Siam-bce | $86.73 \pm 3.71$ | $73.93 \pm 4.98$ | $79.60 \pm 1.94$ | $86.83 \pm 2.45$ |
| EF-wbced | $94.22 \pm 0.85$ | $78.24 \pm 1.72$ | $85.47 \pm 1.06$ | $89.06 \pm 0.86$ |
| Siam-wbced | $95.50 \pm 0.60$ | $82.14 \pm 1.94$ | $88.30 \pm 1.04$ | $91.03 \pm 0.97$ |

Combining the observations made on the scores with the ROC curves seen in Figure 4, it is natural to suspect that the training process of models where the wbced loss function was used features a larger overfitting effect. Indeed, such loss function was originally designed for challenging very-high resolution images, and might not be optimal for Sentinel-2 images. At the same time, using a loss function commonly present in popular deep learning frameworks with the addition of weights inversely proportional to class population seems to provide promising results and can be a choice that better meets the target of this study.

Figure 6 shows the change maps obtained by the four segmentation models for the Beirut city, seen in Figure 1. Change maps obtained by EF-wbced and Siam-wbced confirm the above mentioned suspect of a larger overfitting effect occurring during the training of these models. Indeed, some clusters of changed pixels are detected with high pixel-wise precision, whereas other small changes are not detected at all. More "realistic" results are provided by the models EF-bce and Siam-bce, where changes are detected at the correct locations, but a refinement operation is required in a post-processing phase. The advantage, in this case, is that they can provide better overall results on test data. Furthermore, it is important to remark that the average scores are calculated during the model validation phase and are based on the scores obtained on dataset partitions that still belong to the same original dataset, even though not used at training time.

The goal of the segmentation task was to study the performance of different architectures while working with the same dataset, and observe how a more complex loss function might improve or worsen the results. A test was performed with EF-bce on a completely different dataset, and presented in Section 6.

*5.2. Classification Results*

As previously mentioned, only the EarlyFusion method was adopted in the design of the classification model and the bce function was chosen for training (using weights as for the segmentation models). The model thus developed was named "EF-CNN".

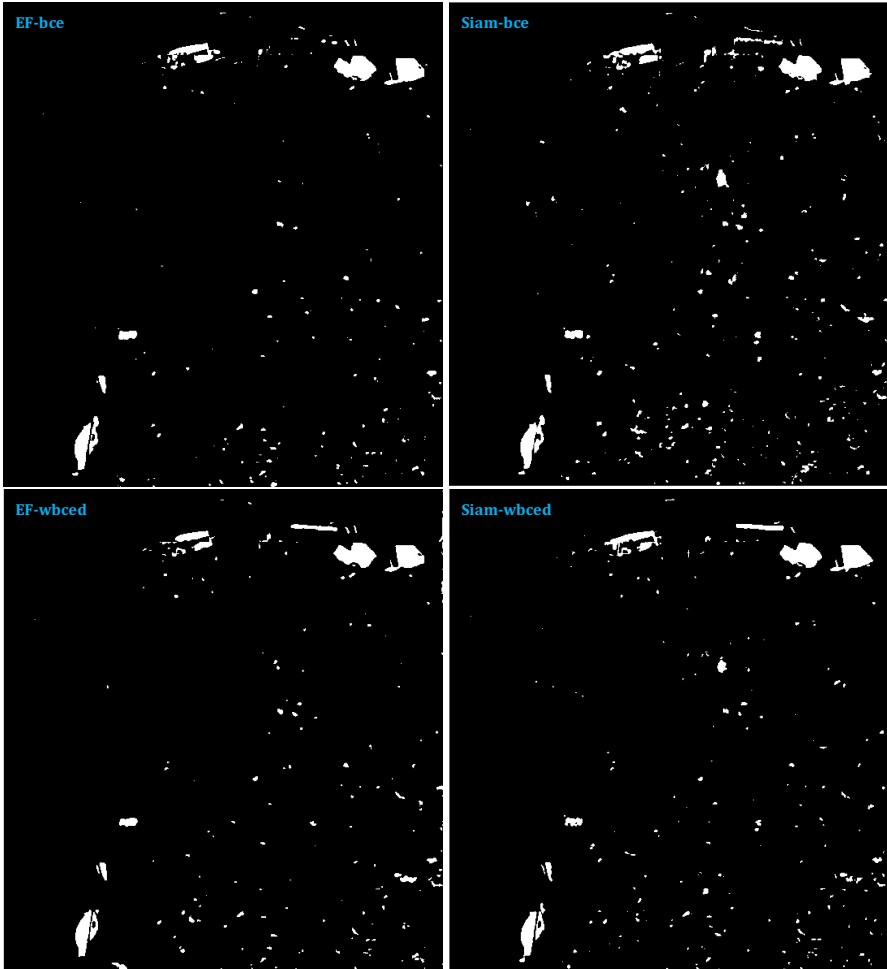

**Figure 6.** Change maps of the Beirut city obtained by the four segmentation models: EF-bce (**top left**), Siam-bce (**top right**), EF-wbced (**bottom left**), and Siam-wbced (**bottom right**).

Figure 7 shows the ROC curves (*left*) of the five models trained with k-fold cross validation and the confusion matrix (*right*) obtained for the classification approach (as before, only the best result is shown). In this case, *AUC* scores are lower with respect to the segmentation task, as expected. Indeed, the ground truth definition might be ambiguous for certain samples, because the image pair was labeled as "changed" if at least 25 pixels hosted changes, but no conditions were set with respect to their distribution across the entire scene (i.e. presence of a large cluster, or changes covering a few pixels each, etc.). Nevertheless, the results are very promising (*AUC* scores still above 85% with the exception of model 4 that might feature a training issue as experienced for the segmentation approach) if considering that the target workflow of this study is to detect possible changes onboard the satellite and, in the positive case, transmit the new acquired image to the ground stations.

In the classification approach, the change threshold was also set to 0.3. In this case, such value refers to the entire image pair and not related to single pixels, as it is used to establish whether the pair contains a considerable number of changes occurred during the period of observation or not.

Table 4 presents the average scores of the classification model and the corresponding uncertainties, calculated as the standard deviations across the five models trained with the k-fold cross validation. Despite the ambiguous definition of the ground truth as previously mentioned, the *recall* is still around the 80% and the *F1-score* is around 77%, thus representing very promising performance for such a simple convolutional neural network.

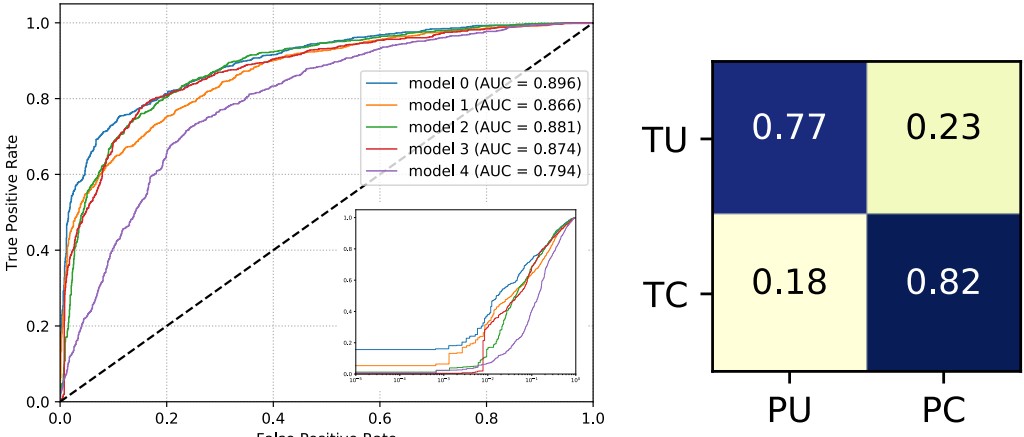

**Figure 7.** (**Left**) ROC curves of the five models trained with the cross validation technique for the classification task. The inset plot shows the ROC curves with logarithmic scale for the *x*-axis. (**Right**) Confusion matrix obtained with the classification approach. The best result achieved with the k-fold cross validation technique is shown for EF-CNN. Rows are true unchanged (TU) and true changed (TC) classes (ground truths), whereas on columns are predicted unchanged (PU) and predicted changed (PC) classes (inference results).

**Table 4.** Average percentage values of the considered metrics for the classification task. Uncertainties are calculated as the standard deviations over the five models trained with the k-fold cross validation.

| Model | Precision | Recall | F1 | Balanced Accuracy |
|---|---|---|---|---|
| EF-CNN | 73.98 ± 5.73 | 80.16 ± 2.77 | 76.72 ± 2.44 | 77.91 ± 3.43 |

An alternative output of the classification model, if not using the change threshold, is a coarse version of the change map. All the pixels of each $128 \times 128$ pixels patch are assigned the output of the CNN for the corresponding image pair, and grayscale is used while combining all the patches back to form the final change map: the darker the patch, the lower the probability of having a considerable amount of changes in that section of the scene. The grayscale change map for the Beirut area is shown in Figure 8. It can be observed how the left border of the map is completely black, as no changes are present in that part of the scene; the $128 \times 128$ pixels patches that cover a large cluster of changed pixels are white, while intermediate situations are generally represented with different levels of gray. This result is generally useful to study how the algorithm is performing on the entire scene, while in the onboard scenario only the CNN output is needed, as it represents the discriminating score that permits to decide whether to transmit or not the new image to the ground station.

Finally, it is important to remark that the classification study is performed assuming that the onboard preprocessing phase includes the operation of co-registration of images acquired on the same region at different times, that represents a possible scenario of future satellite missions. If the image co-registration is not guaranteed onboard, the trained model would show worse performance as the change detection algorithms are usually very sensitive to this kind of issues.

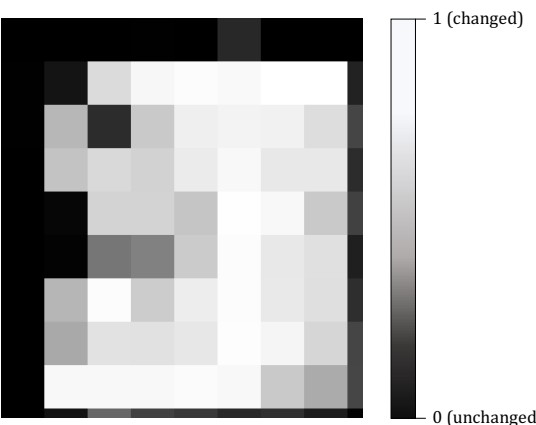

**Figure 8.** Coarse change map of the Beirut city obtained by the classification model EF-CNN. The darker the section of the map, the lower the probability of having a considerable amount of changes in the area.

## 6. Test Case: Rome Fiumicino Airport

The EF-bce model was tested on a new dataset covering another area of interest: the Rome Fiumicino Airport. The ground truth manually detected by photointerpretation for this area was only available for very high-resolution images and, therefore, not all of the labeled changes can be detected on the corresponding Sentinel-2 images, depending on their sizes. Furthermore, not all of the changes present in the ground truth are related to urban variations (the dataset was created for different target studies), while at the same time some urban changes are not labeled at all. The standard approach used for this dataset consists of manual detection of changes by experts in the Earth Observation domain, which grants higher accuracy at the cost of extensive time and human resources. Table 5 shows pros and cons of the standard approach and of the Deep Learning approach here proposed. The period of observation is December 2018–September 2019.

**Table 5.** Pros and cons of two different approaches to change detection with the Rome-Fiumicino Airport dataset.

|  | Pros | Cons |
|---|---|---|
| Standard approach | • High accuracy | • Time consuming (days/weeks)<br>• Not scalable<br>• Depends on human experience<br>• Needs a EO domain expert |
| Deep Learning approach | • Fast<br>• Automatically provides information (also pixel by pixel)<br>• Exploits new technologies (GPUs)<br>• Very low human interaction | • Training phase required<br>• Scarcity of labeled datasets<br>• Risk of overfitting<br>• Fine-tuning of the parameters required |

In order to apply the developed change detection algorithm to this dataset, it was necessary to select only those changes covering an area of 1600 m$^2$ or larger in the ground truths (corresponding to an area of 16 pixels in Sentinel-2 images). Then, after producing the change map with the EF-bce model, a filtering operation was performed by cross-checking the results with available CORINE Land Cover (https://land.copernicus.eu/pan-european/corine-land-cover, accessed on 15 December 2020) maps and Google Earth (https://www.google.com/earth/, accessed on 15 December 2020) images to discard respectively changes in non-urban areas and those along the coast, which could be caused by a larger error in the image co-registration.

A total of 38 changes were detected as passing this filtering phase. While 24 of these changes were confirmed also by human eye, the remaining 14 are $F_P$ and considered particular cases (shadow effects, vegetation changes in the airport area, presence of airplanes

along the runways, etc.). The presence of such a relatively high number of false positives is compatible with the initial decision of detecting as many changes as possible and eventually discard them in the post-processing phase, in order to maximize the *recall* score.

Finally, the 14 changes of the available ground truth that passed the filtering selection about the pixel size were intersected with the changes detected by the algorithm; 5 of them were correctly detected, while 9 are $F_N$. The final results thus consist of 33 true changes, with 24 $T_P$, 9 $F_N$, and 14 $F_P$, for an overall *recall* score of 72.73% and a *precision* value of 63.16%. These results confirm the performance obtained with the OSCD dataset and show that the post-processing phase has great importance in change detection studies to enhance the efficiency of the algorithms. Some of the change detection results are shown in Figures 9 and 10. The images on the left have very high-resolution (the same for the available ground truth), while those on the right are acquired by the Sentinel-2 satellite mission. The red box outlines the changed area detected by the algorithm.

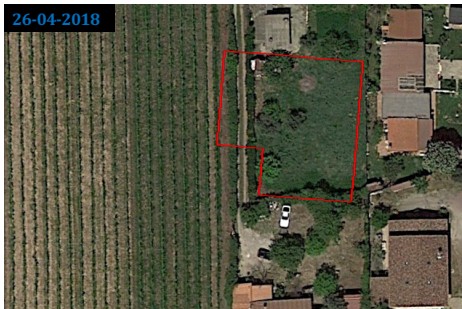 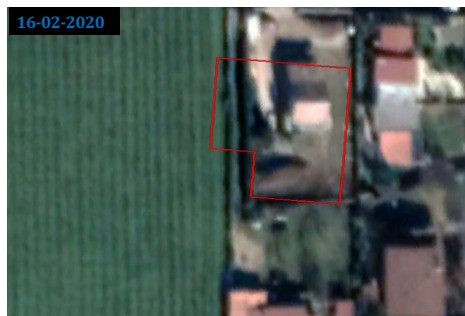

**Figure 9.** Scene comparison of the Rome Fiumicino airport area. The image on the left has very-high resolution, while the image on the right is acquired by the Sentinel-2 satellite mission. The algorithm correctly detects a new house built in the scene.

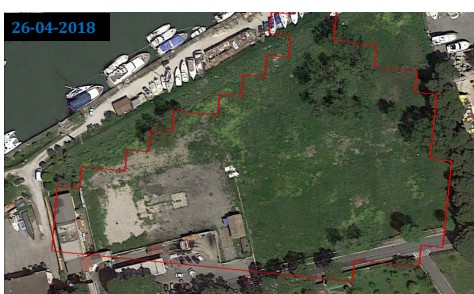 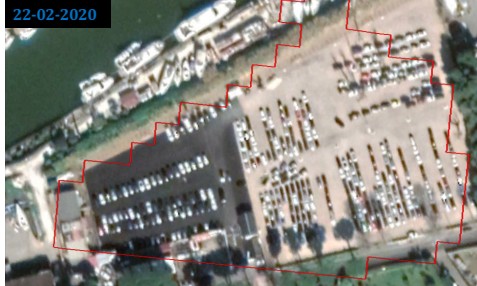

**Figure 10.** Scene comparison of the Rome Fiumicino airport area. The image on the left has very-high resolution, while the image on the right is acquired by the Sentinel-2 satellite mission. The algorithm correctly detects a new parking area in the scene.

## 7. Conclusions

The work presented in this paper shows very promising results that can be considered as a baseline for future studies. Two different architectures were developed for the segmentation approach, while also testing two loss functions: a "classical" binary cross-entropy and its sum with the dice function. In both cases, it was crucial to apply weights in order to face the unbalance problem typical of the change detection datasets. Although the performance of the models is similar, those models trained with the simple binary cross-entropy loss experience a smaller overfitting effect in the case of Sentinel-2 images and should be preferred, thus limiting the choice of the other loss function to the case of very high-resolution images, for which it was originally designed. For the classification task, instead, a very simple CNN model was developed, showing good results that might be improved with a more robust definition of the ground truth. Finally, one of the segmentation models was tested on a new area of observation, where its performance matches

the results obtained with the validation dataset with an expected small loss, that typically occurs with new data.

**Author Contributions:** Conceptualization, A.D.P., N.T., M.I. and D.P.; Formal analysis, A.D.P.; Investigation, A.D.P., N.T., A.P., M.I. and A.D.F.; Methodology, A.D.P. and N.T.; Project administration, A.P. and S.S.; Software, A.D.P., N.T. and A.D.F.; Supervision, A.P. and S.S.; Validation, A.D.P. and N.T.; Writing—original draft, A.D.P.; Writing—review and editing, A.D.P., N.T., A.P., A.D.F. and D.P. All authors have read and agreed to the published version of the manuscript.

**Funding:** This research received no external funding.

**Acknowledgments:** The authors would like to thank the ReCaS (https://www.recas-bari.it/index. php/en/, accessed on 3 October 2021) (a project funded by the Italian Ministry for Education, University and Research in the PON Research and Competitiveness 2007–2013 Notice 254/Ric) data center management team, that provided access to powerful and high-performance devices to support the work here presented.

**Conflicts of Interest:** The authors declare no conflicts of interest.

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
