# Peer review of "Deep Learning Approaches to Earth Observation Change Detection"

_remotesensing, doi:10.3390/rs13204083_

Round 1

Reviewer 1 Report

Review of the manuscript “Deep Learning Approaches to Earth Observation Change Detection”

The manuscript ““Deep Learning Approaches to Earth Observation Change Detection” is well written but poorly formatted. There is no statement explicitly stating what is the aim or the objective of the study. I assume that the objective of the paper is to identify and measure the landscape change using satellite images. considering that this is the objective, then the next observation is that the paper does not have the standard structure, even that is well suited for it. The manuscript does not have Methods, with subsequent Data and Processing methodology, not even Results, as they are called "Performance Study". Even minor, the lack of structure really makes the understanding difficult. Please adjust the architecture of the manuscript according to journal requirements. Second, after reading the manuscript, it is clear that the title is misleading, as it is too broad, and the authors in essence used only CNN. It should be focused and include the main method/finding/ approach of the paper in the title. The paper has two major weakness, even that they are not fatal. First major issue, why the authors focused only on Sentinel-2 and not other satellites, such as Landsat, SPOT or NAIP, which are free for either all the planet or for some areas of the planet (SPOT for Canada). I would suggest including at least one other platform/sensor. Second, there is no information on bands significance, which one is important and which not, considering that an upscaling of some layers was done. Nevertheless, I found the aper suited for publication, once the authors address these two issues and the list of minor comments that I placed at the end of my review, which are focused on improving the discourse.

The detailed comments are identified by line number:

L50: The introduction to Sentinel 2 is too abrupt. What is the connection with the previous paragraph?

L93: eliminate “in order”, as not needed

L94: “128 x128 x13”- what are the units; pixels I assume and for all 13 bands, so say so. If this is the case then 128 pixels for some bands means 1.28 km while for other bands means, like 940 or 1375 nm  means 6 times more. Explanations are needed, as the 60 m cells that were upsampled introduce errors. Consider that 128 x 10 or 128 x 20 does not divide exactly to 60, so the patches have issues, to start with.

L101: a=128, was cropped”- as expected, the units are pixels. For which band the pixels were used for cropping: the ones with 10 m resolution or the ones with 60 m resolution?

L142: “softmax”- please give some explanation /context to the softmax activation.

L148: “the”- eliminate, as typo.

L185: “the loss function”- what is loss function in this context. Explain it, as it means different things in different settings.

L194: “this”- which problem, "apparent high performance"? avoid usage of determiners, as they confuse the reader. state the issue/task/etc, even that it looks unpleasant. This is science, so it is allowed to be coarse, at times.

L225 until line 226: line number is missing.

Eq.4: recall: - I would use the remote sensing terminology beside the standard computer science vocabulary, as omission and commission are wildly used in classification of remote sensing images.

Eq. 6 : F1-score : I would add kappa index, which is more sophisticated than F1, and is widely used in remote sensing.

Author Response

Dear reviewer,

thank you for your useful comments and suggestions, we highly appreciated them.

About the paper structure, we didn't follow the journal format for "Article" as we preferred to keep it in the "Communication" type. Indeed, the aim of the work was to study the performance of two different approaches that exploit CNNs to perform change detection, that might be exploited in more complex workflows by private companies or in academic research.

The reason why we only used Sentinel-2 images is that they are free-access with worldwide coverage (which is important to have a rich dataset) and a high resolution (up to 10m) to detect small changes as well. Other satellite missions were considered, but they either have lower resolution (thus reducing the number of possible changes that can be detected) or not fully coverage of the planet (or at least not for free). In addition, some of the authors work with Sentinel-2 images on other projects, so it was useful to work with this dataset and evaluate the goodness of the results with respect to the current software used. The main idea is to have a sample workflow for Sentinel-2 satellites that could be designed for other future missions as well.

The significance of the bands was studied during the optimization phase of the algorithms. Change predictions obtained with 3 bands (RGB-only), 4 bands (10m resolution  only) and 10 bands (10m + 20m resolutions) were less accurate than with the full 13-bands images. However, an analysis on how much performance loss we could accept to achieve more speed and get less memory occupancy was not made and can be considered for future studies.

Most of the detailed comments have been addressed, and answers for the others are provided below:

L50: added a "bridge"statement at the end of the introduction.

L94: bands at 20m and 60m were upsampled during the creation of the Onera dataset (Ref. Daudt et al.); therefore, the cropping was not influenced by the different resolution.

L142 softmax: we noticed it was a mistake, as we only used sigmoid activation since for the binary classification using softmax is pointless (it's useful in the multiclass case).

Eq.4 and 6: we preferred to use standard computer science terminology, as we found it in most of the remote sensing papers that we consulted. Also, the used metrics were also chosen to evaluate the performance of our models and have a comparison with previous results obtained in the original Onera paper (Ref. Daudt et al.).

Reviewer 2 Report

In the reviewed paper, the authors focused on the deep learning approach to earth observation changes. It is an interesting application and worth investigating. However, some issues should be improved like:
1) The abstract says nothing about novelty, obtained results.
2) Introduction should be extended to state-of-art from the last 3 years top. See the development in machine learning. Discuss also the practical aspects of such approaches which you can see, for example in "Automatic ship classification for a riverside monitoring system using a cascade of artificial intelligence techniques...", where a similar approach (using CNN) was applied and tested in a real environment.
3) Show the features in trained CNN models. 
4) Did you tested your solution on superpixels? I think, better results can be gained.
5) How did you define the CNNs architectures?
6) Some comparisons with state-of-art and other machine learning methods should be made.
7) Compare also your proposal to the commonly known learning transfer architectures.

Author Response

Dear reviewer,

thank you for your useful comments, we highly appreciated them.

Here are the answers to your comments, using the same numbering scheme:

1) Added a few lines to the abstract to better explain the objective/background of the study and the novelty aspect.

2) Added three more references concerning change detection studies in the last few years. We would prefer to keep references related to CD only and not going too far from this task (although there's a quite rich literature about semantic segmentation and scene classification in remote sensing with deep learning techniques). 

3) Attached are some samples from one of the models and for the first two convolutional layers. These feature maps are what one should expect from a CNN that performs analysis on satellite images. Successive layer outputs are not worthy to observe, due to the reduced dimensionality. We preferred not to include this feature maps in the article as they don't really provide additional information to the results, but some of them might be included if necessary. Feedbacks are always welcomed.

4) No, we didn't tested our solution on superpixels yet. They would be, basically, in between the two different approaches that we presented in this article (with some advantages and disadvantages), but they will be certainly considered for future studies, as it is an interesting compromise for our purpose.

5) The CNN architectures were inspired by the UNet paper (Ref. Ronneberger et al.) and the original Onera paper (Ref. Daudt et al.). Several optimizations were certainly made in terms of the main parameters of the networks to achieve good results (number of layers, number of filters, batch normalization layers, etc.)

6)-7) The aim of the study is to have a sample workflow for Sentinel-2 satellites that could be designed for other future missions as well and exploited by both private companies and academic research studies. Therefore, the objective was to study the performance of these algorithms with respect to a standard approach (see Table 5). Using larger and pre-trained networks has generally the additional cost of of worse speed performance and higher memory occupancy.

Reviewer 3 Report

Please see the attached report.

Author Response

Dear reviewer,

thank you for the useful comments, we highly appreciated them.

Here's the answers to your minor comments:

1)

2) The aim of the study is to have a sample workflow for Sentinel-2 satellites that could be designed for other future missions as well and exploited by both private companies and academic research studies. Therefore, the objective was to study the performance of these algorithms with respect to a standard approach (see Table 5) and, at the same time, have a comparison with the previous results obtained in the original Onera paper (Ref. Daudt et al.). Comparisons with other methods in terms of speed, accuracy and memory performance are currently under evaluation and considered for further studies.

3) Table headings placed before the tables. 

Reviewer 4 Report

The article is well structured and well written, with only slight English language editing with minor spell check.

Author Response

Dear reviewer,

thank you for your comments and suggestions.

Best regards,

the authors.

Round 2

Reviewer 1 Report

The second version of the manuscript “Deep Learning Approaches to Earth Observation Change Detection” is a much improved version of the first submission. Minor English improvements are warranted, which I can be addressed during the proofing stage.

Reviewer 2 Report

it can be accepted in the current form

Reviewer 3 Report

I have no further comments.

This manuscript is a resubmission of an earlier submission. The following is a list of the peer review reports and author responses from that submission.

Round 1

Reviewer 1 Report

This paper tackles the problem of change detection in remote sensing images. The authors present two different approaches, semantic segmentation and classification, that both exploit convolutional neural networks.

Two possible methods were evaluated: the EarlyFusion (EF) and the Siamese (Siam) methods. In the EF method, the two images of a pair are concatenated along the channel dimension, and features are extracted from a single image. In the Siam method, feature extraction takes place separately on the two images but uses shared weights between the two paths of the network.

The paper is well-written and presents interesting and promising results in terms of balanced accuracy.

The main concerns are the following:

- The analysis of the literature is quite shallow (12 references) considering that there is a lot of work in this area. Examples include but are not limited to segmentation methods (10.3390/rs13010071, 10.3390/rs13010119) and prediction methods based on modified UNet and generative adversarial networks (see DOI: 10.3390/rs12244142, 10.3390/rs12244145).

- The method proposed is evaluated quantitatively on a single dataset (Table 2). I understand that this depends on the scarcity of labeled data. However, in addition, the models in Table 2 seem to be variations of the proposed method. I would like the authors to clarify this aspect because it would mean that there are no competitor methods considered in the study. Overall, this makes the nature of the study preliminary and raises doubts about its maturity and suitability for a journal publication.  

I suggest the authors prepare a revised version of the manuscript.

Author Response

Dear reviewer,

we would like to thank you for your comments, as they were very useful to improve the quality of the work.

About point 1, we just added 2 more references to remote sensing change detection articles that were explored during this research work. As the article mainly addresses the problem of the change detection, we preferred to limit the references to only papers that handle this task, and avoided references on general semantic segmentation or other methods. Indeed - and this also addresses point 2 - the objective of the study was to explore simple and plan methods (in the specific case, basic convolutional neural networks for segmentation and classification) that can be integrated in more complex onground workflows available at companies providing  remote sensing dedicated services (like Planetek Italia). This also explains why we tested on an "almpost-private" dataset available at the company.  Additional benchmark tests on GPU throughput are being made for a successive publication on the feasibility of the onboard installation of these devices working with such a workflow.

Any additional comment or suggestion is welcome.

Thanks in advance.

Reviewer 2 Report

In this paper, the author presents two different approaches to change detection (semantic segmentation and classification) that both exploit convolutional neural networks to achieve good results, which can be further refined and used in a post-processing workflow for a large variety of applications. 

The paper is written fairly and has good technical content. However, some issues can be addressed to enhance the quality of the manuscript. Hence, I would like the authors to revise the manuscript using the following comments. 

  1. Please add systematic LR Table to show the pros and cons of proposed approach and existing approaches use in the change detection task for both segmentation and classification. 
  2. What are experimental parameter setting values considered? Please construct a Table to show the all parameter setting values for the researcher so they can re-run the experiment in future. 
  3. Why the extracted feature vector size is 128 × 128 × 2? What is the motivation behind this also add reference to support the statement? 
  4. Object diagram of proposed approach is missing 
  5. Pseudocode for proposed approach is missing. 
  6. Figure 3 is not standard figure to show the architecture, please use standard CNN image architecture theme to show the number of layers  
  7. Please compute the time complexity of proposed segmentation and classification approach and compare with recent state of the art algorithms. 
  8. Please perform statistical analysis to validate your obtained results. 
  9. Add confusion matrix to show the participation of each target class in the segmentation and classification.

Author Response

Dear reviewer,

we would like to thank you for your comments, as they were very useful to improve the quality of the work.

Point 1 was addressed with Table 5.

Point 2 was addressed with Table 2.

Point 3 was addressed with an explanation of the choice of 128 as the value for patches size.

For point 4 see point 6.

For point 5 see point 6.

For point 6, we used the same "standard architecture" as in Ref.9 (UNet: Convolutional Networks for Biomedical Image Segmentation), as our models are based on it. Furthermore, this standard provides information about each layer (type of the operation performed and output size), so we believe this format provides all the elements necessary to explain the approach and reproduce the model (addressing points 4 and 5). 

Point 7: the time complexity of a neural network depends on the number of layers, size of the inputs and types of operations performed. In our case, as we are not comparing our algorithms with other ML/DL algorithms but only with the standard approach used on a test dataset (Rome-Fiumicino airport), calculating the time complexity would require additional efforts without providing a metric of comparison with such standard approach (which is much slower as it doesn't use any ML/DL elements/algorithms). In addition, the EarlyFusion and the Siamese methods are very similar in terms of the network architecture and the time complexity would be the same (the difference would be less than an order of magnitude as verified by measuring the inference time).

Point 8: The term "statistical analysis" is a bit generic. For each metric, the mean value and standard deviation are provided and show how statistical fluctuation across the five models trained with the k-fold cross validation technique (for each method/approach) is small. An example (for the CNN approach) is attached. If necessary, this kind of plot can be added in the main body of the paper for each method/approach.

Point 9 was addressed by adding confusion matrices as required for each method/model (Figure 5 and 7).
